# The Effect of the Lysine Acetylation Modification of ClpP on the Virulence of *Vibrio alginolyticus*

**DOI:** 10.3390/molecules29174278

**Published:** 2024-09-09

**Authors:** Shi Wang, Yingying Jiang, Weijie Zhang, Yingzhu Wei, Xing Xiao, Zhiqing Wei, Xiaoxin Wen, Yuhang Dong, Jichang Jian, Na Wang, Huanying Pang

**Affiliations:** 1Fisheries College, Guangdong Ocean University, Zhanjiang 524025, China; 18111025418@163.com (S.W.); 13693494840@163.com (Y.J.); 13687403706@163.com (W.Z.); wyz13984858540@sina.com (Y.W.); 18282553338@163.com (X.X.); 13590993014@163.com (Z.W.); czoteamo@163.com (X.W.); dyh234799@163.com (Y.D.); jianjc@gmail.com (J.J.); 2Guangdong Provincial Key Laboratory of Aquatic Animal Disease Control and Healthy Culture, Zhanjiang 524025, China; 3Chinese Academy of Inspection and Quarantine, Beijing 100176, China; wangna85@yeah.net

**Keywords:** *Vibrio alginolyticus*, ClpP, lysine acetylation, virulence

## Abstract

Acetylation modification has become one of the most popular topics in protein post-translational modification (PTM) research and plays an important role in bacterial virulence. A previous study indicated that the virulence-associated caseinolytic protease proteolytic subunit (ClpP) is acetylated at the K165 site in *Vibrio alginolyticus* strain HY9901, but its regulation regarding the virulence of *V. alginolyticus* is still unknown. We further confirmed that ClpP undergoes lysine acetylation (Kace) modification by immunoprecipitation and Western blot analysis and constructed the complementation strain (C-*clpP*) and site-directed mutagenesis strains including K165Q and K165R. The K165R strain significantly increased biofilm formation at 36 h of incubation, and K165Q significantly decreased biofilm formation at 24 h of incubation. However, the acetylation modification of ClpP did not affect the extracellular protease (ECPase) activity. In addition, we found that the virulence of K165Q was significantly reduced in zebrafish by in vivo injection. To further study the effect of lysine acetylation on the pathogenicity of *V. alginolyticus*, GS cells were infected with four strains, namely HY9901, C-*clpP*, K165Q and K165R. This indicated that the effect of the K165Q strain on cytotoxicity was significantly reduced compared with the wild-type strain, while K165R showed similar levels to the wild-type strain. In summary, the results of this study indicate that the Kace of ClpP is involved in the regulation of the virulence of *V. alginolyticus*.

## 1. Introduction

The post-translational modification (PTM) of proteins is a strategy enabling organisms to effectively control their life activities, which enables them to effectively adapt to environmental changes [1]. Currently, many types of post-translational modification and a large number of PTM sites have been identified, such as glycosylation, phosphorylation, methylation and acetylation [2,3,4]. Among them, protein lysine acetylation, which is dynamic, conserved and reversible, is one of the most extensively studied protein PTMs [5]. It is involved in basic physiological functions such as antibiotic resistance [6], adaptability [7] and virulence [8]. To date, many studies have revealed that acetylation modification is involved in the regulation of bacterial virulence. For example, in *Mycobacterium tuberculosis*, the TcrX protein affects the activity of response-regulated proteins as transcriptional regulators by regulating phosphorylation, and, more importantly, its acetylation mimic K231Q mutant reduces the DNA-binding activity [9], which has important implications in influencing the virulence of *Mycobacterium tuberculosis*. In *Salmonella typhimurium*, the acetylation of PhoP K201 [8] and HilD K297 [10] modulates bacterial virulence by regulating the DNA-binding ability. In addition, the role of acetylation has been reported in fish pathogens. In *Aeromonas hydrophila*, the lysine acetylation K7 site of the AhyI protein not only reduces protease production but also modulates the production of signaling molecules [11]. Zhang et al. showed that that the lysine acetylation sites of AhaI modulate the uptake of several types of antibiotics [12]. Furthermore, it has been shown that there are 1178 proteins with lysine acetylation modifications in the *Vibrio alginolyticus* acetylome profile, of which 102 acetylation-modified proteins are predicted to be virulence factors [13]. However, although lysine acetylation (Kace) modification has been studied in *V. alginolyticus*, the regulatory mechanism between acetylated proteins and virulence is still unclear. 

*V. alginolyticus*, which is a halophilic and warmophilic Gram-negative bacterium, is widely distributed in the aquatic environment [14]. It is a zoonotic pathogen that can infect mariculture animals on a large scale, as well as causing diarrhea, otitis media and septicemia in humans [15,16]. The pathogenicity of *V. alginolyticus* is the result of a combination of virulence factors, including extracellular products, lipopolysaccharides, adhesins and the iron uptake system [17,18,19]. The caseinolytic protease proteolytic subunit (ClpP) is a conserved serine protease that is involved in the transition of various pathogens at different stages to regulate their virulence [20]. The deletion of the *clpP* gene significantly reduces the virulence of *V. alginolyticus* and decreases its adhesion to cytokine-induced killer cells (CIK) [21]. 

In a previous study, the acetylation modification of ClpP at the K165 site was identified by mass spectrometry (MS) [13]. However, the role of the acetylated protein ClpP in regulating the virulence of *V. alginolyticus* requires further study. In the present study, we constructed acetylation modification site mutant strains K165Q (acetylated lysine mimic) and K165R (acetylation-defective) and further evaluated the effect of the acetylated protein ClpP on the virulence of *V. alginolyticus* by determining its extracellular protease activity, LD_50_ and effect on cytotoxicity. Therefore, this study clarifies the regulatory effect of lysine acetylation modification on bacterial virulence and provides a theoretical reference for the prevention and treatment of vibriosis. 

## 2. Results

### 2.1. Characteristics of clpP

The open reading frame of the *clpP* gene was 627 bp, encoding 208 amino acids (accession number: WP_005382265.1), with a predicted molecular mass of 23.04 kDa and a theoretical isoelectric point of 5.09. Multiple sequence alignment showed that the amino acid sequence of ClpP had 92.31–98.08% homology with that of other *Vibrio* species (Figure 1A). It shared the highest homology with the ClpP of *Vibrio parahemolyticus* and *Vibrio campbellii* (98.08%). In addition, as shown in Table 1, the 3D model parameters obtained by SWISS-MODEL showed changes in both the acetylation and deacetylation of ClpP. Visualization analysis showed that the acetylated and deacetylated residue side chains of ClpP were changed (Figure 1B–D). 

### 2.2. Recombinant ClpP Expression, Purification and Antibody Validation

In this study, the prokaryotic expression vector pET-28a-*clpP* was successfully constructed and a recombinant protein with His-tag was successfully expressed. As shown in Figure 2A, a single band of approximately 23.9 kDa (in lanes 5 and 6) was obtained from the purified recombinant protein after SDS-PAGE electrophoresis. In addition, Western blot analysis showed that the serum antibody was able to recognize the ClpP recombinant protein and the band size conformed to 23.9 kDa (Figure 2A, in lane 7), which indicated that the anti-ClpP preparation was successful.

### 2.3. Validation of ClpP Lysine-Acetylated Proteins Using Immunoprecipitation and Western Blot

The Kace of ClpP was further analyzed using IP and Western blotting. As shown in Figure 2B, the molecular weights of the ClpP natural and recombinant proteins were approximately 23.04 kDa and 23.9 kDa, respectively, and they were recognized by anti-ClpP and anti-acetylation antibodies. These results indicate that both the natural and recombinant proteins of ClpP have a lysine acetylation modification.

### 2.4. Growth Curve Measurement

To determine the effect of the site-directed mutant at the K165 site on growth, the growth curves of the wild-type strain of *V. alginolyticus* HY9901, C-*clpP*, K165Q and K165R were determined. As shown in Figure 3A, there was no significant variation in cell density in any of the three strains compared with the WT strain. These results indicate that C-*clpP* and the site-directed mutant strains at the K165 site have no effect on the growth of *V. alginolyticus*. 

### 2.5. Effect of Lysine Acetylation on Biofilm Formation

As shown in Figure 3B, after 24 h of culture, the biofilm formation ability of K165Q was significantly lower than that of the WT strain. In addition, after 36 h of culture, the biofilm formation ability of K165R was significantly increased. However, there was no significant difference in the biofilm formation of the three strains compared witn the WT strain at 12, 48 and 72 h. 

### 2.6. Effect of Site-Directed Mutagenesis Strains on LD_50_ of Zebrafish 

As shown in Table 2, the virulence of K165Q to zebrafish was significantly reduced compared with the WT (*p* < 0.01). However, K165R showed no difference in virulence to zebrafish compared with the WT (*p* > 0.05). The virulence of the complementary strain C-*clpP* was completely restored, almost reaching the level of the WT. Meanwhile, none of the fish in the control group died or appeared to be diseased. 

### 2.7. Effect of Site-Directed Mutagenesis Strains on ECPase Activity

The ECPase results are shown in Figure 4A, which indicates that C-*clpP*, K165Q and K165R had no significant effects on the ECPase activity compared with the WT strain.

### 2.8. Effects of Site-Directed Mutagenesis Strains on GS Cells

GS cells were infected with four strains of bacteria and the lactate dehydrogenase (LDH) activity, nitric oxide (NO) release and glutathione (GSH) content were assayed, evaluating the effects on cytotoxicity. The results showed that the LDH activity was significantly decreased for K165Q compared with the WT-infected group (Figure 4B). Similarly, the NO release was significantly reduced for K165Q (Figure 4C). As shown in Figure 4D, the GSH content of the K165Q group was significantly higher than that of the WT group, and the difference was statistically significant (*p* < 0.01). These results indicate that the effect of the K165Q strain on cells was significantly reduced. 

## 3. Discussion

Bioinformatics analysis plays an important role in the prediction of protein structure and function. In the present study, the results of multiple sequence comparisons showed that the ClpP of *V. alginolyticus* has extremely high homology with that of other Vibrio sequences, which could indicate that ClpP is highly conserved. However, the K165 residue of ClpP is not evolutionarily conserved in Vibrio, probably because some residues of the protein have changed to adapt to the environment [12]. It is possible that mutations in this unconserved region will also affect the regulatory role of the gene. By visualizing and analyzing 3D protein models, we found that the side chains of the residues of K165Q and K165R were significantly changed, especially K165Q. Therefore, the prediction results reveal that such alterations in the spatial structures of proteins caused by site-directed mutagenesis may cause changes in their functions [22]. In addition, the parameters of the protein’s three-dimensional structure model are of great significance in understanding protein function, molecular docking, drug design and so on. In this study, preliminary predictions of the three-dimensional model parameters for the acetylation and deacetylation of ClpP were obtained, and changes in the parameters after acetylation and deacetylation were found. However, the significance of these changes requires further research at a later stage. 

In the mass spectrometry data of a previous study, many lysine acetylation sites were identified in *V. alginolyticus* [13]. ClpP was identified to be acetylated at the K165 site. In order to further verify the acetylation modification of ClpP, we analyzed it by IP and Western blotting. The results showed that the acetylation modification of ClpP was consistent with the previous data. In addition, to better understand the effect of the acetylated protein ClpP on the pathogenicity of *V. alginolyticus*, we successfully constructed site-directed mutant strains. 

Biofilm formation has an important influence on the survival ability of bacteria and the ability to resist the host immune response system. It results in high drug resistance [23,24], anti-phagocytosis [25] and strong adhesion [26], and it is one of the most important causes of bacterial resistance and persistent infection. Recently, it has been found that acetylsalicylic acid (ASA) can reduce the activity of glucosyltransferases (Gtfs), which promote biofilm formation, thereby increasing the level of acetylation and inhibiting *Streptococcus mutans* [27]. Based on previous studies in which lysine acetylation had an important effect on biofilm formation, we measured the changes in the biofilm quantity at different time periods to evaluate their effects on *V. alginolyticus* biofilm formation. In our study, the acetylation of ClpP at the K165 site (K165Q mutant strain) was significantly lower than the biofilm formation level of the WT strain at 24 h of incubation, which indicates that the acetylation of ClpP inhibits biofilm formation in *V. alginolyticus* to some extent. Interestingly, when incubated for 36 h, we found that the K165R mutant strain had significantly higher biofilm levels compared with the WT strain, while the K165Q strain showed similar biofilm levels to the WT strain. These results are preliminary in nature, as only biofilms at a single point in time were examined. However, the acetylation and deacetylation of the AhyI protein in *Aeromonas hydrophila* did not affect biofilm formation [11]. In contrast to the results of this study, the reasons may be closely related to the different strains and different protein functions. 

The extracellular proteases of Vibrio have critical functions and are considered as potential virulence factors [28]. The study of Chen et al. showed that the deletion of *clpP* did not affect the ECPase activity of *V. alginolyticus* [21], which is similar to our results. In our study, C-*clpP*, K165Q and K165R exhibited no significant difference in ECPase activity compared with the WT strain, suggesting that the acetylation modification of ClpP does not affect the expression of extracellular proteases. However, in *V. alginolyticus*, lysine deacetylation at the K52 and K68 sites of PykF significantly reduced the ECPase activity, whereas deacetylation at the K317 site showed no significant effect on ECPase activity [29]. 

In *Salmonella typhimurium*, when compared with the deacetylation of K297, the acetylation of HilD at the K297 site has been reported to attenuate virulence in a mouse model [10]. This result suggests that the deacetylation of K297 plays an important role in *Salmonella*’s virulence, which is similar to this study. We used each strain to infect the zebrafish model, and the results showed that the virulence of K165Q to zebrafish was significantly reduced compared with the WT strain, while the K65R strain exhibited no significant change in virulence to zebrafish. However, compared with K165Q, the virulence of K165R in the zebrafish model was significantly increased. These results suggest that the lysine acetylation of ClpP plays an essential role in *V. alginolyticus*. Interestingly, we found that although K165R enhanced biofilm formation, it showed no difference in the infected zebrafish model. Similarities have also been found in Liu’s research [30]. However, the specific reasons need to be further studied. 

LDH is a stable protein found in the cytoplasm of normal cells. When the cell membrane is damaged, it is released from the cell. The quantitative analysis of cytotoxicity can be achieved by detecting the LDH activity released from cell-membrane-ruptured cells into the culture medium [31], which has been widely used in various bacteria [32,33,34]. In our study, GS cells were infected with four strains to determine the LDH release, which showed that the LDH release of the K165Q group was significantly lower than that of the WT strain group. Therefore, this result could indicate that the effect of the K165Q strain on cytotoxicity was weakened.

NO release is considered an important indicator used to assess cytotoxicity [35,36,37]. The enzyme that catalyzes the synthesis of NO is called NO synthase (NOS) [38]; at present, it is divided into three types, namely endothelial NOS (eNOS), neuronal NOS (nNOS) and inducible NOS (iNOS). In addition, the overexpression or imbalance of iNOS can produce high concentrations of NO, which can lead to toxic effects [39]. Using GS cells infected with each strain, we determined the NO release. The results showed that the value in the K165Q group was significantly lower than that for the WT strain, while there was no difference between the K165R and WT strain groups. Thus, these results suggest that the acetylation of the K165 site reduced the virulence of *V. alginolyticus*. 

Glutathione (GSH), which is an endogenous antioxidant necessary for the maintenance of cellular function and proliferation [40], is widely found in human and animal tissue cells. The depletion of GSH is considered to be one of the mechanisms associated with cell apoptosis [35]. Moreover, studies [41,42,43] have shown that high levels of GSH can increase the resistance to oxidative stress and the antioxidant capacity. However, the lack of GSH or changes in the GSH/GSSG ratio can cause cells to become more susceptible to oxidative stress, inflammation and other undesirable states. In this study, the GSH content of GS cells infected with the K165Q strain was higher than that of the WT strain group, and the difference was significant. However, the GSH content of the K165R strain infection group showed a similar level to that of the WT infection group. In other words, compared with the K165Q-infected group, the WT and K165R groups exhibited a reduced GSH synthesis capacity of GS cells, which could be attributed to the depletion of GSH, leading to the elevation of ROS or an imbalance in the normal ratio of GSH/ROS [44], resulting in cell cycle arrest or even induced cell death. However, this requires further exploration and research. We can confirm that the acetylation of the ClpP protein at the K165 site is involved in the regulation of the virulence of *V. alginolyticus* and reduces its pathogenicity. This is of great significance in further exploring the role of lysine acetylation modification in fish pathogens. 

## 4. Materials and Methods

### 4.1. Bacterial Strains and Fish 

*V. alginolyticus* wild-type strain HY9901 was isolated from *Lutjanus erythopterus* [45]. The Δ*clpP* mutant was constructed by Chen et al., and the deletion of the *clpP* gene did not affect its normal growth [21]. *V. alginolyticus* was cultured at 28 °C in tryptic soy broth (TSB, Haling, Shanghai, China). *Escherichia coli* strains were cultured with Luria–Bertani (LB, Huankai Co., Ltd., Guangzhou, China) at 37 °C. Zebrafish (average weight 0.35 ± 0.05 g and average length 3 ± 0.05 cm) from a commercial fish farm in Zhanjiang were identified as healthy via bacteriological recovery experiments. The zebrafish were temporarily reared in a water recirculation system at 28 °C for two weeks prior to the experiment. The primers used during the research are listed in Appendix A. The appropriate amount of antibiotics was added according to the requirements: ampicillin (Amp, 100 μg mL^−1^); kanamycin (Km, 50 μg mL^−1^); chloramphenicol (Cm, 25 μg mL^−1^). 

### 4.2. Cloning and Bioinformatics Analysis of clpP Gene from V. alginolyticus HY9901

A pair of primers, *clpP*-F and *clpP*-R, was designed based on the *V. alginolyticus* genome sequence (accession number: CP072781-CP072782) and used to amplify the gene using the *V. alginolyticus* strain HY9901 as a template. PCR was performed in a Thermocycler (Bio-Rad, Hercules, CA, USA) under the following optimized amplification conditions: pre-denaturation at 95 °C for 5 min, followed by 33 cycles of 95 °C for 30 s, 58 °C for 30 s and 72 °C for 30 s. After the amplification products were detected by agarose gel electrophoresis, the products were recovered from the agarose gel to ligate into the pMD18-T vector (TaKaRa, Dalian, China) and transformed into *E. coli* DH5α (Transgen, Beijing, China). The positive clone was selected and sequenced by Sangon Biological Engineering Technology & Services Co., Ltd. (Shanghai, China). The similarity between the nucleotide sequence and the deduced amino acid sequence was analyzed via the BLAST program and the protein was analyzed with the ExPASy tool [46]. SWISS-MODEL (https://swissmodel.expasy.org) was used to build 3D protein models, which were then visualized and analyzed using UCSF Chimera 1.17.3. 

### 4.3. Prokaryotic Expression of clpP and Antibody Preparation 

To construct the expression vector, a pair of primers, pET-28a-*clpP*-F and pET-28a-*clpP*-R (containing EcoR I and Xho I sites at the 5′ and 3′ ends, respectively), was designed. *V. alginolyticus* strain HY9901 was used as a template for PCR and detected by agarose gel electrophoresis. The product was then recovered from the agarose gel. The product was linked to the double enzymatically cleaved pET-28a vector according to the instructions of the Ready-to-Use Seamless Cloning Kit (Sangon Biotech, Shanghai, China). Following this, the recombinant product was transformed into *E. coli* BL21 (Transgen, Beijing, China). Positive clones were selected (primers: pET-28a-YZ-F/R) and sequenced to confirm the successful construction of the expression vectors. The transformants were grown at 37 °C in Luria–Bertani (LB, Huankai Co., Ltd., Guangzhou, China) with kanamycin (50 μg/mL). When these cultures reached an OD600 = 0.5–0.6, 1 mM IPTG (1:100) was added to induce protein expression for 8 h at 28 °C. After induction, the bacteria were washed twice with 1× phosphate-buffered solution (PBS, pH 7.4) and collected by centrifugation at 8000× *g* for 5 min. Suspended bacteria were then placed on ice and broken using an ultrasonic homogenizer (Scientz, Ningbo, China) at power of 40%. The resulting cell lysate was centrifuged at 10,000× *g* for 20 min to collect the clarified supernatant. In addition, bacteria (1 mL) were washed with PBS and collected; they were then boiled with 50 µL of 1 × buffer for 5 min to obtain the whole bacterial protein. Protein purification was performed using the method of Sheng et al. [47]. Briefly, the His-tagged proteins were washed and collected with different concentrations of imidazole (10, 20, 50, 75, 100, 150, 200 mM). Each sample was analyzed by 10% reducing SDS-PAGE.

The antibody was prepared from the purified ClpP protein. Mice were immunized according to the instructions for the QuickAntibody-Mouse3W (Biodragin, Suzhou, China). Briefly, the ClpP recombinant protein was used as an antigen and was rapidly mixed with adjuvants in a 1:1 volume ratio under sterile conditions, followed by injecting the immunized SPF-grade mice through the calf muscle of the hind leg at 100 μL per mouse. After two injections (each injection contained 20 µg protein) in three weeks (on the 21st day), the blood of the mice was collected and placed at 4 °C overnight. Then, it was centrifuged at 3000× *g* to collect the serum to obtain anti-ClpP. The protein was quantified according to the Bradford Protein Assay Kit (Beyotime, Shanghai, China) instructions. The success of antibody preparation was analyzed by Western blotting. 

### 4.4. Immunoprecipitation and Western Blot

To verify that the ClpP protein had acetylation modifications, we extracted the natural protein of ClpP from *Vibrio alginolyticus*. In short, the natural protein of ClpP was obtained by performing the IP method of Zeng et al. [48]. Lysates (700 μL) of *V. alginolyticus* strains were interacted with anti-ClpP (1 μL) at 4 °C and left overnight. The cell lysates were then transferred to a Protein A Agarose column, washed with 1 × PBS (pH 7.4) and incubated at 4 °C for 3h. After this, the supernatant was discarded and the agarose beads were washed 6 times with 700 µL of IP buffer and collected. Finally, 50 μL buffer was added and the mixture was boiled for 5 min and analyzed by SDS-PAGE and Western blotting. The Western blot procedure was performed according to Ren et al. [49]. Briefly, 20 µL of protein was run on a 10% 1-DE gel and transferred to a polyvinylidene fluoride (PVDF, Millipore, Burlington, MA, USA) membrane. This membrane was infiltrated with the QuickBlock™ Blocking Buffer (Beyotime, Shanghai, China) for 15 min at room temperature. The primary antibodies used in the Western blot were anti-ClpP and anti- acetylation mouse mAb at a 1:2000 dilution (PTM Biolabs Inc., Hangzhou, China), followed by two hours of incubation. Horseradish peroxidase (HRP)-conjugated goat anti-mouse IgG as the secondary antibody (1:10,000) was used and it was incubated for 1 h. Finally, the membrane was visualized using an ECL system (Bio-Rad, Hercules, CA, USA), and an automatic chemiluminescence image analysis system (Tanon 5200) was used to obtain photographs of the experimental results [50]. 

### 4.5. Construction of clpP Complementation Strain (C-clpP) and Acetylation Site-Directed Mutagenesis Strains

The *clpP* gene promoter sequence was analyzed using the Promoter 2.0 website (http://www.cbs.dtu.dk/services/Promoter/) (accessed on 10 April 2023), and primers (*clpP*-com-F, *clpP*-com-R) with two sites, Pst I and Xho I, were designed according to the pBBR1MCS-1 plasmid map. *V. alginolyticus* strain HY9901 was used as a template and the PCR product was recovered after a PCR procedure and detection by agarose gel electrophoresis. Then, the product was ligated into the pBBR1MCS-1 vector, which was transformed into *E. coli* DH5α, and positive clones were selected. The correctly sequenced strain was used to extract the recombinant plasmid named pBBR-*clpP* and it was stored at −20 °C until use. Meanwhile, the recombinant plasmid was transformed into *E. coli* S17-1λpir and bound to Δ*clpP* via a bacterial conjugation assay. Briefly, the S17-1λpir- *clpP* and Δ*clpP* strains were mixed in a 3:1 ratio, and the mixed strains were coated in trypticase soy agar (TSA, Huankai Co., Ltd., Guangzhou, China) medium overnight for 18 h. Following this, the strains were rinsed with 1 mL of TSB liquid medium and collected, and the bacterial solution was diluted in a 10-fold concentration gradient. Then, 100 μL of the bacterial solution was coated on thiosulfate citrate bile salts sucrose agar culture medium (TCBS, Huankai Co., Ltd., Guangzhou, China) containing chloramphenicol and cultured at 28 °C for 2 to 3 days. Finally, sequencing was performed to determine the successful construction of C-*clpP*. 

A fast mutagenesis system kit (Transgen, Beijing, China) was used to construct site-directed mutant strains including K165Q (lysine to glutamine, mimic acetylation) and K165R (lysine to arginine, mimic deacetylation). Primers (K165Q-*clpP*-F and K165Q-*clpP*-R, K165R-*clpP*-F and K165R-*clpP*-R) were designed using the website (https://crm.vazyme.com/cetool/singlepoint.html, accessed on 15 June 2023). Moreover, the pBBR1-*clpP* mentioned above was used as a template to amplify the target plasmid with reference to the kit instructions. The amplification products were digested by DMT restriction endonuclease at 37 °C for one hour (degradation of methylated plasmid templates in plasmids to achieve higher mutation efficiency). The product was then transformed into *E. coli* S17-1λpir and sequenced. It was combined with Δ*clpP* via a bacterial conjugation assay and sequenced to determine the successful construction of the site-mutant strains. 

### 4.6. Growth Curve of Bacteria

Bacterial growth curves were obtained according to the method of Shi et al. [51]. Wild-type strain HY9901, C-*clpP*, K165Q and K165R (OD_600_ = 0.5) were inoculated in fresh TSB medium at a ratio of 1:100 and incubated at 28 °C for 24 h. Every 2 h, we measured the OD_600_ and set up three replicates of the experiment. The growth curve was finally drawn by taking the average value. 

### 4.7. Quantification of Biofilm Biomass

The quantification of the biofilm biomass was assayed with reference to the method of Yang et al., with minor modifications [52]. Briefly, the bacterial inoculum (OD_600_ = 0.5) was diluted at a ratio of 1:10, separately inoculated into 96-well cell culture plates (150 µL of bacterial solution per well), and incubated at 28 °C for 12, 24, 36, 48, and 72 h. After this, the suspended cells and medium were discarded and washed twice with sterile 150 µL 1 × PBS (pH 7.4). The biofilm was fixed with 150 µL anhydrous methanol, dried at room temperature, and stained with 1% crystal violet (150 µL per well). Then, excess stain was washed off and the specimen was dried at room temperature before being dissolved through the addition of 95% ethanol, and the biofilm was quantified at 570 nm. 

### 4.8. Fifty Percent Lethal Dose (LD_50_)

The LD_50_ values of *V. alginolyticus* strain HY9901 (WT), C-*clpP*, and the site-directed mutagenesis strains (K165Q and K165R) were evaluated to assess the virulence to healthy zebrafish. The injection concentrations were 10^4^, 10^5^, 10^6^, 10^7^, and 10^8^ CFU/mL, respectively. Each fish was injected with 5 μL by intramuscular injection in the experimental group (10 fish for each group), while the control group was injected with 5 μL sterile PBS. The fish were observed for 7 days until the mortality rate stabilized. The experiment was duplicated 3 times, and the LD_50_ values were calculated via the approach of Reed and Muench [53].

### 4.9. Detection of Extracellular Protease (ECPase) Activity

Extracellular protease (ECPase) activity was assessed according to the method of Zhang [54]. Briefly, each bacterial solution (1 mL, OD_600_ = 0.5) was spread individually on TSA solid plates lined with sterile cellophane and incubated at 28 °C for 24 h. The bacteria were then rinsed with PBS and the supernatant was collected by centrifugation at a low temperature. The protease activity of the supernatant was determined using the azocasein trichloroacetic acid colorimetric method at OD_422_. The experiment was set up with three replicates. 

### 4.10. GS Cell Cytotoxicity Assay 

The grouper spleen (GS) cells used in this study were as previously described [55]. They were cultured in Leibovitz’s L15 medium containing 10% fetal bovine serum (FBS, Gibco, Waltham, MA, USA) [56]. Then, 1 mL of cell suspension was added to each well of a 12-well cell culture plate and 200 µL of 1 × 10^6^ bacterial solution, which was suspended in L15 medium, was added to the adherent cells. Then, the cell culture medium and adherent cells were collected after 2.5 h. The content was determined at OD_550_ and OD_450_, respectively, according to the instructions of the nitric oxide (NO) and lactate dehydrogenase (LDH) kits (Nanjing Jiancheng Bioengineering Institute, Nanjing, China). Briefly, the nitrate reductase enzyme is used to specifically reduce NO_3_^−^ to NO_2_^−^, and the concentration is determined by the shade of the color. Lactate dehydrogenase (LDH) catalyzes the formation of pyruvate from lactic acid, which reacts with 2,4-dinitrophenylhydrazine to form dinitrophenylhydrazone pyruvate, which is brownish red in an alkaline solution, and the enzyme activity can then be determined by colorimetry. The content of glutathione was determined at 412 nm according to the GSH and GSSG detection kits (Beyotime, Shanghai, China). All of the above experimental procedures were repeated 3 times. 

### 4.11. Statistical Analysis

The experimental data were analyzed with the SPPS 19.0 software package (SPSS Inc., Chicago, IL, USA). Statistical analyses were performed using one-way ANOVA and Student’s *t*-test to determine significant differences between the groups. *p* < 0.01 (**) indicated highly significant differences and *p* < 0.05 (*) indicated significant differences compared with the control group. 

## 5. Conclusions

This study elucidates the effect of ClpP on the virulence of *V. alginolyticus* from the perspective of lysine acetylation modification. We demonstrated that deacetylation at the K165 site significantly increased the level of biofilm formation, while acetylation at this site significantly decreased the level of biofilm formation. In addition, through LD_50_ and cell infection experiments, the results revealed that the acetylation modification at the K165 site was involved in the virulence regulation of *V. alginolyticus* and reduced its pathogenicity. In summary, the results of this study are important in further exploring the role of lysine acetylation modifications in fish pathogens.

## Figures and Tables

**Figure 1 molecules-29-04278-f001:**
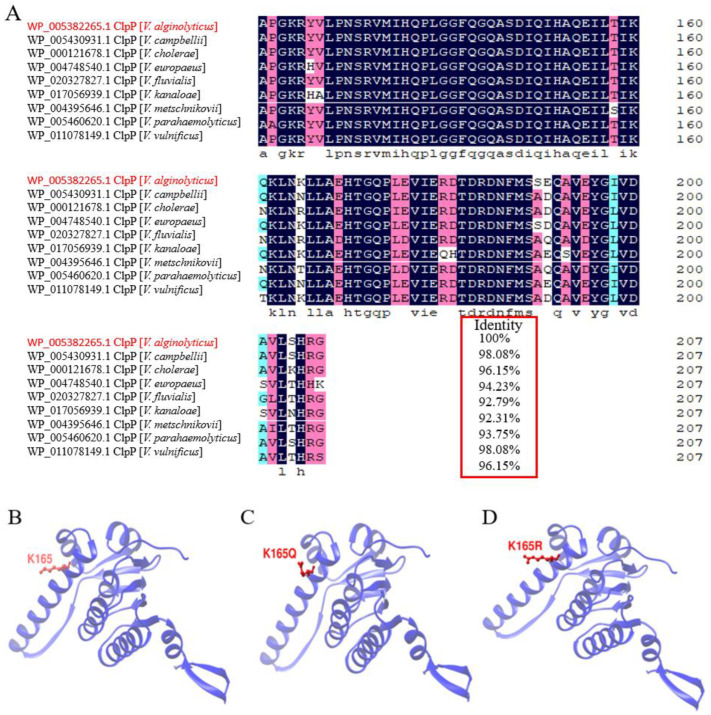
(**A**) Multiple sequence alignment of ClpP from different *Vibrio* species. *Vibrio alginolyticus* ClpP is highlighted in red. The degree of consistency of ClpP with other sequences is indicated by the values in the red box. (**B**) The 3D visualized protein model of ClpP at the K165 site in *V. alginolyticus* HY9901. The K165 active residue is indicated in red. (**C**) The 3D visualized protein model of the acetylation of ClpP at the K165 site. The predicted acetylated residue is represented in red. (**D**) The 3D visualized protein model of the deacetylation of ClpP at the K165 site. The predicted deacetylated residue is highlighted in red.

**Figure 2 molecules-29-04278-f002:**
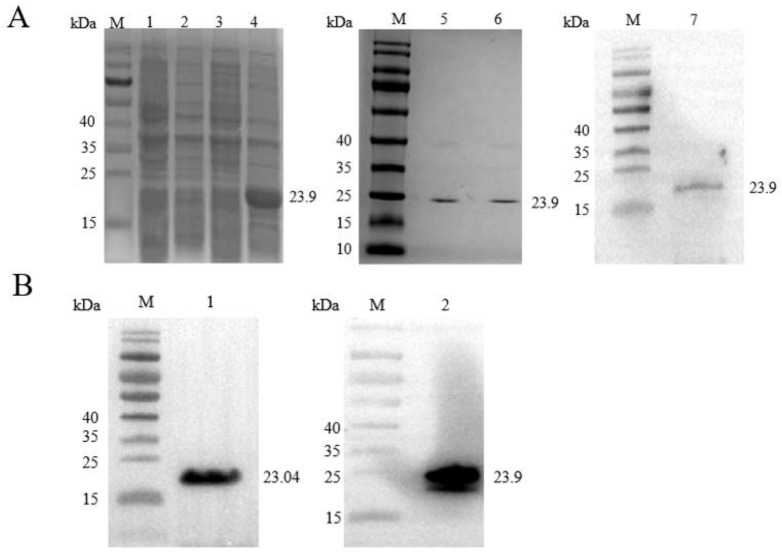
(**A**) SDS-PAGE of ClpP and Western blot of anti-ClpP. The uninduced whole bacterial protein of ClpP is shown in lane 1, the IPTG-induced whole bacterial protein of ClpP is shown in lane 2, the uninduced supernatant protein after the crushing of ClpP is shown in lane 3, the IPTG-induced supernatant protein after the crushing of ClpP is shown in lane 4, the purified supernatant protein of ClpP is shown in lanes 5 and 6 and the Western blot result of the anti-ClpP antibody is shown in lane 7. (**B**) The lysine acetylation modification of the ClpP natural protein and recombinant protein was identified by immunoprecipitation and Western blotting. The ClpP natural protein was enriched by IP with a specific antibody (anti-ClpP), followed by Western blot with the ClpP protein-specific antibody (in lane 1) and Western blot with anti-acetylation mouse mAb (in lane 2).

**Figure 3 molecules-29-04278-f003:**
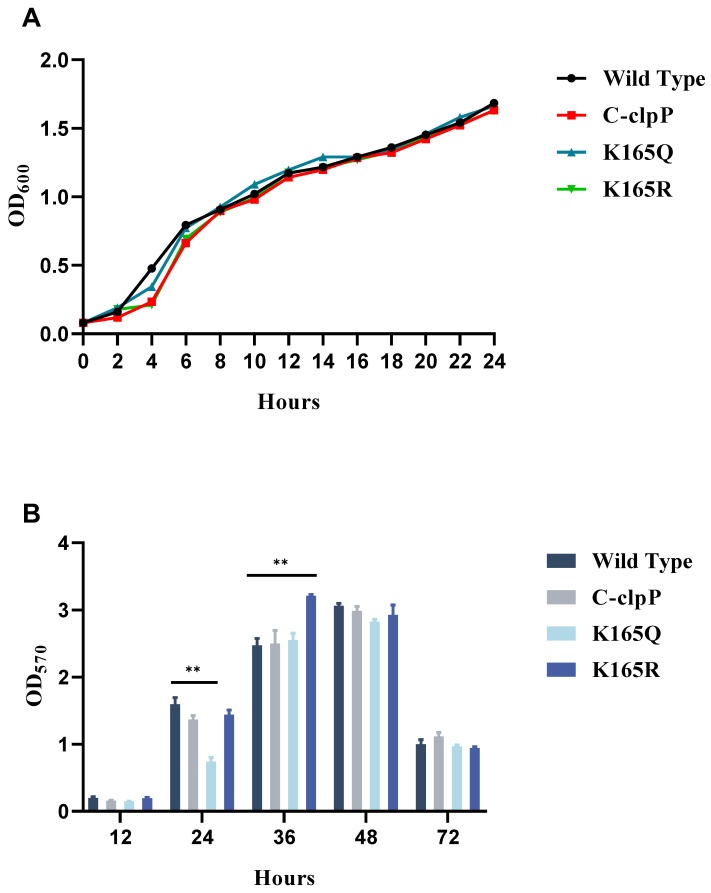
(**A**) Effect of lysine acetylation on the growth of *V. alginolyticus*. (**B**) Effects of C-*clpP*, K165Q and K165R on the biofilm formation of *V. alginolyticus* at different time points. The average SD was obtained from three independent experiments. ** *p* < 0.01, significantly different compared with the WT strain.

**Figure 4 molecules-29-04278-f004:**
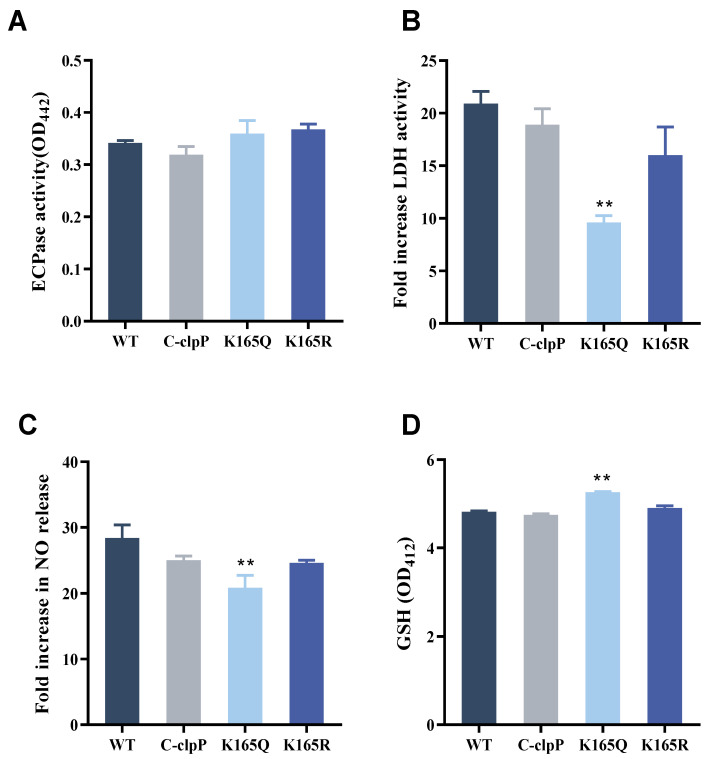
(**A**) The histogram of the effect of the lysine acetylation of the ClpP protein on the ECPase activity of *V. alginolyticus*. All values are mean ± SD, n = 3. (**B**) Effect of lysine acetylation on LDH activity in GS cells. (**C**) Effect of lysine acetylation on NO release in GS cells. (**D**) Effect of lysine acetylation on GSH content in GS cells. The average SD was obtained from three independent experiments. ** *p* < 0.01, significantly different compared with the WT strain.

**Table 1 molecules-29-04278-t001:** The 3D protein structural model parameters of ClpP, K165Q and K165R.

Strain	QMEAN	Cβ	All Atom	Solvation	Torsion
ClpP	1.14	1.03	1.03	1.88	0.32
K165Q	1.26	0.98	1.10	1.88	0.44
K165R	1.14	1.03	1.01	1.93	0.31

**Table 2 molecules-29-04278-t002:** The values of the LD_50_ for the WT, C-*clpP*, K165Q and K165R.

	WT	C-*clpP*	K165Q	K165R
LD_50_ (cfu/mL)	5.82 × 10^6^	5.89 × 10^6^	9.75 × 10^7^ **	7.34 × 10^6^

Values are averaged from three independent experiments. ** *p* < 0.01, significantly different compared with the WT strain.

## Data Availability

All data generated during this study are provided in the manuscript.

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
