# Peer review of "The Effect of the Lysine Acetylation Modification of ClpP on the Virulence of Vibrio alginolyticus"

_molecules, 2024, doi:10.3390/molecules29174278_

Round 1

Reviewer 1 Report

Comments and Suggestions for Authors

The reading of scientific research must be clear enough, informative, learning-prone, enjoyable. I found this paper interesting, and it demonstrated a considerable amount of work. However, I also found it difficult to read, hard to follow, not engaging, and challenging to understand. 

Reading the paper I found the following imppressions:

-        The introduction can provide more insight, leading to understanding and enjoying the paper

-        Methods can be hardly reproducible and highly confusing. A methodology is supposed to be understood and reproducible by anyone.

-        There is a lack of information in the text

-        The writing style is unclear

-        The paper is written in a way that does not allow a smooth and comprehensive approach. 

-        Authors may consider that science and research results can be communicated clearly, helping to engage and attract readers and other researchers.

-        Keep in mind that acronyms must be defined the first time they appear in each and every section, figure, or table. This is because a reader can start reading at any point

 Comments about the paper:

INTRODUCTION

-        After the first “Post-translational modification”, please indicate the acronym PTM.

-        Aim: “In the present study, we constructed acetylation modification site mutant strains K165Q (acetylated-lysine mimic) and K165R (acetylation defective) and further evaluated the effect of the acetylated protein ClpP on the virulence of V. alginolyticus by determining its extracellular protease activity, LD50, and effect on cytotoxicity”

MATERIALS AND METHODS

-        Section 2.1 

o   Please be specific when you said, “Constant aeration was maintained at 28°C for two weeks before experiment”. Does this sentence relate to zebra fish maintenance?

o   Please improve the sentence. For example: "The primers used during the research are listed in ...."; For example: "The appropriate amount of antibiotics was added according to the need..."

-        Section 2.2: 

o   Please enounce the conditions of the PCR amplification. Template? (Genomic DNA, extraction method, cDNA, annotated gene number?)

o   The authors followed the method of Pang et al.; however, it is recommended to briefly describe the method for better comprehension. A reader may not be interested in reading that research or may not have access to it.

o   It is unclear the reference number 12 (Zhang, et al. 2022) referred for the SWISS-MODEL, which is found here https://swissmodel.expasy.org

-        Section 2.3: 

o   Please specify which fragment was obtained or it was a gene? And using which template? under what conditions? Ligation process or Gibson assembly? Please be clearer in your processes.

o   Expression conditions: medium volume, induction temperature, agitation speed.

o   PBS stands for? (All acronyms must be defined the first time of their appearance). Buffer concentration? Buffer pH?

o   The author said: “IT was washed….”. But it is not clear what was washed. I supposed it was cells. Please specify the sonication conditions and how the authors got the soluble proteins. It is unclear the sentence “boiled to obtain whole bacterial protein”. What did you mean? You boiled the soluble faction? Or you boiled to perform the SDS-PAGE?

o   Please specify the purification method, at least, briefly (ion exchange, affinity, SEC?)

o   Please specify the method used for protein quantification

o   The immunization protocol is unclear and hardly reproducible by anyone else: adjuvant used, protein concentration, number of inoculations, etc. According to other reports, the time to obtain antibodies (3 weeks) was short.

o   Please specify the protocol of antibody purification and purity assessment. Did the authors used complete serum? Please be clear.

-        Section 2.4:

o   Please explain how you extracted the natural protein from V. alginolyticus.

o   Authors have specified the ClpP acetylation in a previous work (reference 22); Why do you have to do it again? 

o   Why are you talking about detect “modifications”? Probably, authors must support with experimental data, the idea of detecting modifications in proteins using Complex-Immunoprecipitation.

o   Co-IP and western-blot methods must be described at least briefly (conditions, buffers, resins, immobilization, etc.)

o   It is unclear if antibody anti-ClpP was obtained by mice immunization or obtained from PTM Biolabs, as the antibody anti-acetylation mouse. Both antibodies were used at 1:2000? How did the authors determined de amount of produced ClpP antibody to be used in the assays?

o   Please write in a clearer and sharp way

-        Section 2.5:

o   It may be helpful to understand the rationale behind the analysis of the clpP promotor. It is unclear for a reader

o   The mentioned primers were used to amplify which fragment, and from which template. Unclear methodology

o   The procedure used to obtain the clpP complementation strain is insufficient, and therefore unclear. Any methodology must be reproducible. A reader must understand the procedure. And please be specific indicating the aim to construct

o   In section 2.1, the authors mentioned “ΔclpP mutant was constructed by Chen” and it seems, according to the writing, that it is a V. alginolyticus strain. Now, in section 2.5 the authors wrote “the recombinant plasmid was transformed into E. coli S17−1λpir and bound to ΔclpP by bacterial conjugation assay”. Therefore, it is not clear where is ΔclpP., in a bacterium or in V. alginolyticus. Please be clearer and it would be helpful to clarify why you wanted to construct C-clpP.

o   What happened with the plasmid obtained in section 2.2? Why did the authors construct another plasmid with the clpP gen?

o   Where can be found the tool “CE Design software”?

o   What does it mean DMT? All acronyms must be clearer defined.

o   Please specify clearly, the bacterial conjugation assay.

-        Section 2.7

o   Volume of bacterial inoculum in each well?

o   PBS pH and concentration?

o   Volume of crystal violet (1%) per well?

o   The authors reported that the biofilm was quantified by absorbance. However, it is unclear. What was exactly quantified after the ethanol addition? Please explain the principle of the method.

o    Which controls did the authors use? Did the authors construct a calibration curve? Considering the absorbance reading, a calibration curve must be established.

-        Section 2.8

o   The authors must explain the concentration difference in the CUF used to each strain

o   Explain briefly the approach of Reed and Muench

o   Unclear methodology; hard to reproduce

-        Section 2.9

o   What does it mean TSA? Acronyms must be defined the first time they appeared

o   It is unclear when the procedure explicit the following “They were washed…” (plates, cellophane?)

-        Section 2.10

o   Please describe the principle of the nitric oxide (NO), lactate dehydrogenase (LDH) and GSH and GSSG detection methods. Perhaps they are part of a kit, but there must be a principle that can be explained and then the reader can understand the process.

RESULTS

-        Section 3.1

o   Please specify the source of the information related to the open reading frame of the clpP gene (NCBI, accession number?)

o   As a reader, I still do not know how you got the gene.

o   Software(s) used to perform multiple sequence alignment

o   The 3D computational models show a very similar structure in the three ClpP proteins (WT, K165Q and K165R). The difference in only an amino acid lateral chain. A computational model cannot be considered as a strong conclusion of different structures of proteins that also were analyzed using the same template. Authors must indicate the protein used to construct the models (PDB, Uniprot?). Probably, the lateral chain is important to explain the difference in the protein activity, however, deep studies must be conducted to assure that the structure has changed in a degree that explain a possible activity. 

o   Please improve the image quality of Figure 1, it is a little blurry.

o   In table 1, please explain the meaning of “QMean” or “Cß”. Are those results presented in the table enough to assure a difference among structures? Which is the significance of the results? Which was the angle considered to report the “torsion” value? It may be recommended to show a deeper information of the results presented in the table. The table title says, “D structural”, is this right? if it so, what does it mean?

-        Section 3.2

o   I keep thinking that the methodology is not enough clear to demonstrate expression, purification, and detection of ClpP protein. Did the authors use any control? (positive or negative). 

o   Figure 2. The figure explanation is not clear enough. Figure 2A: Why did the authors show two supernatant proteins? Which is the aim of two bands with the same sample? (this is unclear). The western blot shows a band and little lower than the band in the SDS-PAGE; maybe a control would be helpful. Figure 2B: Is “ClpP lysine-acetylated protein” the same ClpP protein? Why are the authors naming the protein in a different way? The sentence: “ClpP protein was enriched by Co-IP with specific antibody (anti- ClpP)” is unclear and it was not presented in the methodology (in section 2.4 it was presented the purification of ClpP protein using Co-IP, but not enriched). What do the authors mean by “anti-lysine acetylation antibody”? In section 2.4 says “anti-acetylation mouse”, it is the same?

-        Section 3.4

o   The results from growth curve of all strains, can may be just mentioned. I consider thar the figure is unnecessary. It does not give relevant information. 

-        Section 3.6

o   The information contained in the table is already in the section text. It is unnecessary to repeat information. Please choose only one way to show your results.

o   Please specify the units of the results. Remember that a reader may begin to read in any section of a paper or just read tables and figures, so the information must be clear anywhere.

o   How can the authors demonstrate that all the mutagenesis were effective? The author can sequence a plasmid, but how to know that the V. alginolyticus was effectively mutated?

-        Discussion

o   3D protein models cannot clearly support the following sentence: “we found that the spatial structures of K165Q and K165R were greatly altered, especially K165Q”. There is a change in the lateral chain of only one residue. Additionally, that is a computational prediction, the authors cannot really know the conformational until a X-ray or RMN is performed

o   It is unclear the relation between acetylsalicylic acid (ASA) and this research (“Recently it has been found that acetylsalicylic acid (ASA) can acetylate proteins via a non-enzymatic transacetylation reaction and thus have an inhibitory effect on Streptococcus mutans [39]. Moreover, ASA reduces the activity of glucosyltransferases (Gtfs), which promote biofilm formation, and increases the level of acetylation”)

o   Perhaps the K165Q mutant strain produced lower biofilm at 24 hours, but the biofilm production was the same at other times in the experiment. Therefore, there is not a clear relevance of the mutant. Maybe produce a little change which is then restored in time.

Comments on the Quality of English Language

.

Author Response

Thanks for your comment. Please see the attachment.

Reviewer 2 Report

Comments and Suggestions for Authors

The virulence-associated caseinolytic protease ClpP is acetylated at K165 in Vibrio alginolyticus HY9901 but its effect on regulation of the virulence of V. alginolyticus is still unknown. The authors conclude that lysine acetylation of ClpP is involved in regulation of V. alginolyticus virulence.

Unfortunately I find too many inconsistencies and I cannot recommend publication. Some examples follow:

A multiple alignment is shown in Fig 1A. the proteins have 92-98% homology, yet the amino acid they are interested in, K165, is not identical. Four sequences have K at the site, two have R, two have Q, and one has T. If this was an important site for lysine acetylation, then the K residue should be conserved. Instead only 4 out of 9 sites have a lysine. In fact two have a Q residue, which should mimic the fully acetylated lysine, and two have an R residue, which should mimic the fully unacetylated lysine. One might conclude from this that lysine acetylation at K165 is not likely to have an important role in clpP regulation that is conserved across all of these strains. This should be discussed.

Figure 1 B, C, and D show 3D visualizations of K165, K165Q and K165R proteins. These show very little difference in structure, as you would expect for mutations of a residue with a side chain that is exposed to the solvent. These differences could be within the margin of error as the proteins undergo their molecular and rotational vibrations. Table 1 does not indicate anything interesting or useful about the structures of the different proteins. As a result, these data do not show, as claimed in the text, that the protein conformation will likely be altered by acetylation. If anything, protein acetylation would instead be expected to alter the protein affinity for DNA or for interaction with another protein in vivo.

Figure 2: “Co-IP” is an indirect immunoprecipitation method that identifies protein-protein interactions. An antibody to protein 1 will indirectly immunoprecipitate protein 2 if there is a protein-protein interaction between proteins 1 and 2. I believe the technique used here was IP, not Co-IP. If that is incorrect, then the technique needs to be described in more detail.

Figure 2A, lane 3: why are the size markers visible? Are they fluorescent? This should be described.

Figure 2B, lane 1 shows the natural protein, while lane 2 shows the recombinant protein. What is the purpose here? Is the second band in lane 2 supposed to show acetylation? I doubt that acetylation of K165 would cause such a large shift in mobility seen in lane 2. Point mutations of a charged residue does not show a large mobility shifts like that. You would need a control showing the mobility shift is specifically due to acetylation, and not due to proteolysis, for example. Or that the K165Q and R mutants are not recognized by the anti-lysine acetylation antibody.

Figure 3A: Why not show the growth curve of the delta-clpP strain as a control to show the effect of the absence of the clpP gene on cell growth?

Figure 3B: The observation of less biofilm at the 24-hr time point would be an important conclusion. However, one time point is not enough by itself to show that acetylation of ClpP inhibits biofilm formation here. You would want to see that the effect is supported by similar results at nearby time points such as 22 and 26 hr. I would want to see that the 24-hr result would not disappear with the additional time points.

Figure 3B: Similarly, you would want to show that the 36-hr result with the K165R mutant is not an isolated event, but that it could also be seen at 34 or 38 hr to support the 36-hr result.

Table 2: For the WT and C-clpP strains, we expect that the protein is acetylated in the cell. So the K165Q result should mimic those results, and show no difference from them. The K165R mutant cannot be acetylated and should mimic the result of a LACK of acetylation of ClpP. Instead, the fact that K165Q shows a big effect and K165R shows no effect is strong evidence that acetylation of K165 does not occur to any significant extent in the conditions of this experiment. However, it shows that if K165 was acetylated, then there would be a big effect. So that could be very interesting for understanding ClpP function. If I have misunderstood, and I have this wrong, then I apologize, but in that case, the discussion of this experiment needs to be expanded so the reader can understand it better.

Overall, the Introduction and the Discussion are well-written, However, in the Results section, the conclusions about the role of ClpP acetylation need to be re-examined and the section needs to be given a major re-write.

Round 2

Reviewer 1 Report

Comments and Suggestions for Authors

The authors replied neatly to all comments, and I appreciate their efforts. 

Thank you.

I just have a few other issues:

-        At the end of the first paragraph of section 4.3, the authors repeatedly used the word “collect or collected.” Please improve the writing

-        Sometimes, the authors used “ClpP” and others “clpp”. Please unify the writing.

-        Section 4.4: define the IP acronym

Comments on the Quality of English Language

.
